# Terahertz Biosensor Engineering Based on Quasi-BIC Metasurface with Ultrasensitive Detection

**DOI:** 10.3390/nano14090799

**Published:** 2024-05-04

**Authors:** Jun Peng, Xian Lin, Xiaona Yan, Xin Yan, Xiaofei Hu, Haiyun Yao, Lanju Liang, Guohong Ma

**Affiliations:** 1Department of Physics, Shanghai University, Shanghai 200444, China; 18816925990@163.com (J.P.); linxian01@shu.edu.cn (X.L.); xnyan@shu.edu.cn (X.Y.); 2School of Opto-Electronic Engineering, Zaozhuang University, Zaozhuang 277160, China; yxllj68@126.com (X.Y.); hxf1010sk@163.com (X.H.)

**Keywords:** terahertz, metasurfaces, sensors

## Abstract

Terahertz (THz) sensors have attracted great attention in the biological field due to their nondestructive and contact-free biochemical samples. Recently, the concept of a quasi-bound state in the continuum (QBIC) has gained significant attention in designing biosensors with ultrahigh sensitivity. QBIC-based metasurfaces (MSs) achieve excellent performance in various applications, including sensing, optical switching, and laser, providing a reliable platform for biomaterial sensors with terahertz radiation. In this study, a structure-engineered THz MS consisting of a “double C” array has been designed, in which an asymmetry parameter α is introduced into the structure by changing the length of one subunit; the Q-factor of the QBIC device can be optimized by engineering the asymmetry parameter α. Theoretical calculation with coupling equations can well reproduce the THz transmission spectra of the designed THz QBIC MS obtained from the numerical simulation. Experimentally, we adopt an MS with α = 0.44 for testing arginine molecules. The experimental results show that different concentrations of arginine molecules lead to significant transmission changes near QBIC resonant frequencies, and the amplitude change is shown to be 16 times higher than that of the classical dipole resonance. The direct limit of detection for arginine molecules on the QBIC MS reaches 0.36 ng/mL. This work provides a new way to realize rapid, accurate, and nondestructive sensing of trace molecules and has potential application in biomaterial detection.

## 1. Introduction

Terahertz (THz) waves are far infrared electromagnetic radiation located between microwaves and infrared waves, which have been paid considerable attention in a wide range of fields, such as materials science, chemistry, and biomedicine, due to having large penetration depth, being nonionizing and nondestructive, and for the identification of molecular fingerprints [1,2,3,4]. However, the development of THz devices is limited by relatively long wavelengths and a lack of natural materials to respond effectively to THz waves. The emergence of metamaterials offers an alternative strategy to solve this problem. Metasurface (MS) refers to artificial composite structures or composite materials with extraordinary physical properties that natural materials do not have. The periodic arrangement of metamaterials makes them have extraordinary properties [5,6,7,8], which has attracted increasing attention from the communities of physics, materials science, photonics, and optoelectronics. In the past decades, abundant kinds of metamaterials have been proposed and designed. Apart from the conventional metallic metamaterials, advanced metamaterials, such as photonic crystals [9,10], all-dielectric metamaterials [11,12,13], flexible metamaterials [14,15], liquid crystal metamaterials [16,17], and so on, are designed and fabricated to provide a potential way to implement ultrasensitive sensors in the THz region [18,19,20,21,22]. The presence of largely localized electric field enhancement near the resonance frequency is the dominant mechanism for the high sensitivity of THz MSs. It is found that the sensitivity of THz metamaterial sensors can be effectively improved by combining nanogold particles or graphene layers with Ms. For example, a metallic ring-shaped THz metamaterial sensor combined with Au nanoparticles was proposed in [23] to further increase the localized electric field around resonance frequency, and the hybrid device can achieve a detection limit (LoD) of 4.2 femtomolar for sensing SARS-CoV-2 viral proteins. Shi et al. designed a fully dielectric THz subsurface combined with functionalized gold nanoparticles [24]. Functionalized Au nanoparticles can greatly improve the sensitivity of the sensor. The highest detection sensitivity can reach 2.96 GHz·mL/nmol, which is 2.66 times that without Au nanoparticles, and the sensor can realize the specific immunoassay of human influenza hemagglutinin tag protein. Lee et al. [25] used the electro-optical characteristics of graphene-bonded nanogap MSs at terahertz to identify very low concentrations of single-stranded deoxyribonucleic acid and found that the performance of terahertz nanosensing tools can be significantly improved by adding graphene layers. The above studies all improve the performance of THz sensors by combining additional materials (nanogold particles and graphene layers) with MSs. Here, we hope to improve the performance of THz sensors by studying the mode resonance of MSs and obtain a THz sensor with a simpler and more sensitive structure.

Classical dipole MS THz biosensors are generally less sensitive due to their inherently low Q-factor [26] resonance patterns and further attenuation by ohmic or radiation loss, hindering the development of THz biosensors. Fortunately, the concept of bound state in the continuum (BIC) has been introduced into the design of MSs in recent years [27,28]. BIC represents special resonance modes with frequencies within the radiative continuum domain but completely bound without any energy leakage [29,30,31]. Theoretically, a BIC has nonradiative characteristics with disappearing spectral linewidth, which can achieve ultrahigh or even unlimited radiation Q-factor and unlimited lifetime [32,33]. Through resonant coupling or breaking the structural symmetry, BIC can produce limited leakage and develop a quasi-bound state in the continuum (QBIC) [34,35,36]. The concept of QBIC has been widely used in the design of optical cavities and highly sensitive sensors due to narrow linewidth and high Q-factor [37,38], such as harmonic generation and biological chemical sensing [39,40,41,42]. In general, there are two design strategies for the high performance of QBIC-based THz sensors. One is the mode-coupling-type QBIC, for instance, a THz MS composed of a split resonant ring [43]; by adjusting the size of the split ring gap, the LC resonance can couple with the dipole resonance, and a BIC with a high Q-factor is obtained. The other is the symmetry-broken-type QBIC, i.e., by introducing structural perturbations into an MS supporting a BIC, QBIC emerges in the far-field response [44].

In this paper, we propose a double C-shaped metal MS that supports QBIC resonances with a highly localized electric field, in which QBIC resonant frequency is tunable with asymmetry degree, the α of the MS structure. BIC, as well as QBIC MS, has been designed and investigated systematically via simulation, coupled harmonic model calculation, and experimental demonstration. The results show that the designed QBIC devices show very high sensitivity for detecting concentrations of arginine biomolecules: the amplitude of the resonance based on the QBIC mode variation is 16 times larger than that of the classical dipole resonance. The LoD for sensing the arginine molecule can reach 0.36 ng/mL based on the designed QBIC MS.

## 2. Materials and Methods

To achieve the BIC resonance mode with a high Q-factor, we designed a metal MS, as illustrated in Figure 1a,b. It consists of an array of double C-shaped metal structures periodically arranged in a square lattice, as shown in Figure 1a. The metal MS, buffer layer, and substrate are chosen as aluminum, polyimide (PI), and fused silica (SiO_2_), respectively. The periodicity of MS along the x and y directions is set to be Px = Py = 140 μm, respectively. The thickness of the Al, PI, and substrate are 2, 10, and 300 μm. As illustrated in Figure 1b, the parameters of the unit are L1 = 90 μm, w = 15 μm, h = 20 μm. Here, the length L2 of the structure is variable to introduce asymmetry, while other parameters remain unchanged. We define the asymmetry parameter α = ΔL/L1 with ΔL = L1 − L2. The simulation is performed by the frequency domain solver of CST Microwave Studio (2023), which was designed by “CST-Computer Simulation Technology AG” company in Frankfurt, Germany. A periodical boundary condition is set in the x–y plane, and floquet mode is selected for numerical simulation. The incident light is set to be in the form of a plane wave, the electric field is polarized along the y direction, and it is normally incident on the MS along the z direction. 

The proposed MS was fabricated on the top of 10 μm PI film supported by a double-sided polished SiO_2_ substrate. Initially, the PI film with a thickness of 10 μm was coated on a 300 μm SiO_2_ substrate; after that, a 200 nm aluminum film was deposited on the top of PI by the magnon sputtering method. Subsequently, photolithography and peeling-off processes were performed on the Al film to obtain the periodical “double C” structure. During simulation, the dielectric constant and tangent loss of PI were treated with 3.1 and 0.05, and the Al is a lossy metal in CST material library, with a conductivity of 3.56 × 10^7^ S/m. The optical image of the fabricated structure is shown in Figure 1c, and Figure 1d shows the schematic diagram of the QBIC MS sensing process. The reagents containing sample molecules are brought into contact with the biosensor surface by dropping the solution. After drying, the sample molecules will be freely distributed and attached to the biosensor surface in the form of small clusters. During the experiment, a commercial THz spectrometer (Advantest TAS7400, DVANTEST corporation, Tokyo, Japan) was used to measure the THz transmission, The effective THz spectral range is 0.5–4.0 THz, with a spectral resolution of 7.6 GHz. Experiments and tests were conducted at room temperature and in a dry nitrogen atmosphere with a humidity of less than 3%. All samples were tested multiple times to ensure the accuracy of the results.

## 3. Results and Discussions

To analyze the mechanism of the symmetry-protected BIC, we introduced an asymmetry degree, α, by changing L2 in the metal superstructure, and calculated the THz transmission spectra under different α values, as presented in Figure 2a, from which we can clearly see the evolution from BIC mode to QBIC mode with the change in α. When α = 0, the BIC of the structure takes place around 0.64 THz, and the BIC is marked with a virtual coil, in which the structure shows an infinite Q-factor, and the energy is trapped without leaking into free space. As α increases, the resonance absorption with narrow linewidth gradually appears, the absorption magnitude increases gradually, and the peaking absorption position shows blueshift with increasing the magnitude of α. This originates from the fact that when α ≠ 0, the structural symmetry is broken, and the symmetry-protected BIC state begins to transform into the QBIC state with a limited leakage rate. Figure 2b plots the degree of the asymmetric α-dependent Q-factor of the proposed structure. The Q-factor was calculated with *Q* = *ω*_0_/(2*γ*), where ω_0_ and γ are the resonant frequency and damping rate, respectively. When α approaches 0, the Q-factor of the QBIC MS approaches infinity. For the two structures with L2 equal to 100 μm and 80 μm respectively, they have equivalent |α = ±0.11|, they all have almost the same Q-factor, and the dependence of the Q-factor on α also follows an inverse quadratic of α. The Figure of Merit (FoM) is introduced here to obtain the best compromise between Q-factor and resonance strength, and here, FoM=Q×I with *I =*
$\left|T_{max}-T_{min}\right|$, in which *T_max_* and *T_min_* correspond to the maximum and minimum transmission of the MS. Thus, the optimal performance of the MS under different α is evaluated. The calculated FoM as a function of α is presented in Figure 2c, showing that the magnitude of FoM in the QBIC increases with |α|. For instance, FoM ≈ 0 for α = 0, while FoM = 8 for α = 0.11, as marked with a blue dashed line; the line width of the dipole resonance becomes narrow with increasing the magnitude of α (higher frequency side in Figure 2a), which means that breaking the structural symmetry also improves the quality factor of the dipole resonance. As shown in Figure 2d, the changes in the resonance response of the dipole mode and the QBIC mode under different degrees of asymmetric α can be observed. BIC is achieved when α approaches zero (star position), in which the resonance transmission peak disappears. It is obvious that the introduction of asymmetry breaks the symmetry-protected BIC state, and THz transmission evolves into a QBIC resonance with extremely narrow line widths. As the degree of asymmetry of α increases further, the QBIC resonant linewidth shows a trend from narrow to wide.

For the BIC mode, the energy transmission of electromagnetic waves is mainly bound in the *z*-axis direction, that is, the Poynting vectors are all in the *z*-axis direction, which also means that the magnetic field direction is on the x–o–y plane. By contrast, for QBIC mode, the Poynting vector is no longer limited to the *z*-axis direction due to energy leakage, which causes a part of the magnetic field to be generated outside the x–o–y plane. Therefore, monitoring the magnetic field component in the z direction clearly shows the energy leakage of THz waves. To further understand the formation mechanism of QBIC, we monitored the magnetic field intensity and surface current along the z direction at 0.64 THz and 0.66 THz (corresponding to the frequencies of the BIC and QBIC modes with α = 0 and α = 0.11, respectively). Figure 3a,d shows the magnetic field and surface current diagram of the BIC state at 0.64 THz for the case of α = 0. Here, there is no energy leakage at the symmetrical protected BIC position, in which the resonance Q-factor is infinite. Figure 3b,e shows the magnetic field and surface current diagram of the QBIC state at 0.66 THz for the case of α = 0.11. The magnetic field of the QBIC mode is stronger than that of the BIC mode, indicating that there is obvious magnetic field leakage, and a sharp transmission valley appears under this condition. Figure 3c shows the electric field enhancement of the MS BIC response (α = 0) and quasi-BIC response (α = 0.11). Figure 3f shows color plots of the electric field enhancement of the MS with varying α and frequency. The MS that supports QBIC mode provides large local field enhancement due to the high FoM factor. Figure 3c,f shows that a QBIC with α = 0.11 achieves a maximum electric field enhancement of about 160 at the resonant frequency in mixed mode. At the BIC position (marked by pink stars), the electric-field-enhanced resonance disappears because the incident wave cannot couple into the metasurface’s hybrid mode.

In order to further reveal the basic physics and formation conditions of QBIC resonance mode excitation in the metasurface, we calculate the electric and magnetic field distribution along the x, y, and z directions at 0.66 THz (QBIC position), respectively. It can be clearly seen from Figure 4a–f that the electric (magnetic) component of the surface plasmon mode propagating in the x–y plane is mainly distributed along the y(z) direction, because the surface plasmon resonance shows the same electric field direction as the induced electric moment in the metasurface. When the surface plasmon mode grazes the surface of metamaterial, it can interact with the dipole mode in the metasurface. Such a diffraction coupling can strongly suppress radiative damping, and because the electromagnetic field of the surface plasmon mode is captured, the QBIC mode is formed at 0.66 THz.

In the following, we employ a universal coupling mode to further analyze the THz transmission spectra of QBIC MSs with varying α. According to coupling mode theory [45,46]:(1)$$\left[\begin{array}{cc} \omega -\omega_{a}-{i\gamma }_{a} & \Omega \\ \Omega & \omega -\omega_{b}-{i\gamma }_{b} \end{array}\right]\left[\begin{array}{c} a\\ b \end{array}\right]=\left[\begin{array}{c} \sqrt{\gamma_{a}}E\\ \sqrt{\gamma_{b}}e^{i\phi }E \end{array}\right]$$
where ω is the frequency of input THz wave. ω*_a_* and ω*_b_* correspond to the resonant frequencies of different modes. Ω is the lossy coupling strength, with Ω = *g* − *i*$\sqrt{\gamma_{a}\gamma_{b}}e^{i\phi }$, where *g* is the coupling strength between two metamaterial structures. γ*_a_* and γ*_b_* are the loss terms of the structures, respectively. For a single metamaterial structure, they are close to γ_1_ and γ_2_ (γ*_a_* = γ_1_; γ*_b_* = (γ_1_ − γ_2_)/2). ϕ represents the phase difference that is calculated with ϕ = (ϕ_1_ − ϕ_2_) *d*, where ϕ_1_ and ϕ_2_ are the phases of the individual C-shape in one unit structure, respectively. The d and E represent the width of the metamaterial structure and the electric field of the incident THz wave, respectively.

Let *a* and *b* represent the electric amplitude stored in the individual C-shape of one unit, respectively, which can be obtained by solving Equation (1), which reads
(2)$$a=\frac{\left(\left(\omega -\omega_{b}-{i\gamma }_{b}\right)\sqrt{\gamma_{a}}-\Omega \sqrt{\gamma_{b}}e^{i\phi }\right)E}{\left(\omega -\omega_{b}-{i\gamma }_{b}\right)\left(\omega -\omega_{a}-{i\gamma }_{a}\right)-\Omega^{2}};$$
(3)$$b=\frac{\left(\left(\omega -\omega_{a}-{i\gamma }_{a}\right)\sqrt{\gamma_{b}}-\Omega \sqrt{\gamma_{a}}e^{i\phi }\right)E}{\left(\omega -\omega_{b}-{i\gamma }_{b}\right)\left(\omega -\omega_{a}-{i\gamma }_{a}\right)-\Omega^{2}}.$$

Subsequently, we calculate the effective susceptibility $\chi_{eff}$ that is the linear superposition of the amplitudes *a* and *b*. The effective electric susceptibility of the metamaterial can be written as follows [47]:(4)$$\chi_{eff}=\frac{\sqrt{\gamma_{a}}a+\sqrt{\gamma_{b}}e^{i\phi }b}{\epsilon_{0}E}$$

Finally, we obtain the transmission spectrum with $T\approx 1-Im\left(\chi_{eff}\right)$, which reads as follows:(5)$$T\approx 1-Im(\frac{\left(\omega -\omega_{a}-{i\gamma }_{a}\right)\gamma_{b}e^{2i\phi }+\left(\omega -\omega_{b}-{i\gamma }_{b}\right)\gamma_{a}-2\Omega \sqrt{\Upsilon_{a}\Upsilon_{b}}e^{i\Phi }}{\left(\omega -\omega_{b}-{i\gamma }_{b}\right)\left(\omega -\omega_{a}-{i\gamma }_{a}\right)-\Omega^{2}})$$

Figure 5a shows the calculated THz transmission spectra of QBICs with Equation (5) for α = −0.11, 0, 0.11, 0.22, 0.33, and 0.44. By comparing Figure 2a and Figure 5a, the calculated THz transmission shows good agreement with that of the simulation. When α = 0, BIC appears at about 0.64 THz, and the BIC is marked with a dashed coil. The resonance has an infinite Q-factor. After the symmetry is broken, the QBIC mode is excited, accompanied with the gradual emergence of the THz transmission valley, from which it is clearly seen that the absorption magnitude increases, accompanying a frequency blueshift with increasing asymmetry degree α. Figure 5b presents 2D plots of THz transmission with respect to frequency and asymmetry degrees α. The BIC position is marked with a blue star, where it is in a symmetrical protection state and there is no energy leakage.

We then apply the proposed QBIC MS for detecting arginine molecules. Figure 6a–e presents the transmission spectra of the THz MS with different α values. The black dashed line is the experimental data, and the red solid line is the simulated curves. Figure 6a shows the case of α = 0, in which the structure has C2 symmetry and there is no energy leakage, and the transmission spectrum lies in the BIC state. Figure 6b–e shows the THz transmission spectra of QBIC MS with α = 0.11, 0.22, 0.33, 0.44, in which the QBIC resonance peak is clearly observed due to the breaking of structural symmetry. Basically, the simulations are consistent with the experimental data for various values of α. The experimental results show a certain deviation from the simulated results, especially for the case of small α. This is due to the fact that (1) during the lithography process, the edges of the unit structure are blurred due to the optical diffraction effect, so that the pattern on the photoresist is not completely consistent with the pattern on the reticle; (2) the inevitable ohmic, scattering, and impurity losses caused by materials, impurities, and rough surfaces during sample preparation; and (3) the limited resolution THz spectrometer used makes it impossible to observe narrow QBIC line shapes. A large deviation is clearly observed in Figure 6b because QBIC MS with α = 0.11 has a very narrow resonance line shape that is beyond the resolution of the THz spectrometer. However, for the case of α = 0.44 in Figure 6e, the resonance line width of the QBIC MS is wide enough that it can be well distinguished with our THz spectrometer, and clearly the measured THz transmission spectrum shows a good agreement with the simulated curve. We adopted the device with α = 0.44 for sensing arginine molecules in the following section.

Figure 7 presents the comparison of THz transmission spectra for a classical dipole MS (α = 0) and QBIC MS with α = 0.44 for sensing arginine molecules, in which the Q-factor of the QBIC under test is ~10. Arginine is an essential precursor for the synthesis of a variety of molecules in the human body, which can participate in the regulation of body growth and development, immune response, and intestinal health and other physiological processes, and it helps correct the acid–base balance of hepatic encephalopathy; therefore, the detection of arginine concentration is of great significance for the diagnosis and treatment of related clinical diseases. We dilute 25 μL of arginine stock solution with a concentration of 1.0 mg/mL into 1.0 mL of distilled water. After that, we add 25 μL of the diluted solution into another 1.0 mL of distilled water to obtain a more diluted solution. We repeat the operation eight times, and we can obtain eight kinds of arginine solutions with concentrations of 24.4 μg/mL, 595 ng/mL, 14.5 ng/mL, 0.35 ng/mL, 3.0 pg/mL, 0.21 pg/mL, 5.12 fg/mL, and 0.12 fg/mL. We take 15 μL of dilute solution and drop it on the QBIC metasurface to form a circular area with a diameter of approximately 5 mm. We continue to drop cumulatively different concentrations of microorganisms onto the QBIC MS. After drying, the sample molecules will be freely distributed and attached to the biosensor surface in the form of small clusters, as shown in the inset in Figure 7d. Finally, we selected the arginine biomolecules with different concentrations (arginine concentrations were 0, 0.36, 1.07, 15.57 and 44.57 ng/mL) on classical dipole MSs and QBIC MSs for comparison, which are shown in Figure 7a,b, respectively. It can be found that as the concentration of arginine biomolecules increases, the transmitted amplitude of both the classical dipole MS and the QBIC MS change decrease. Figure 7d,e presents the enlarged views of the amplitude changes in the classical dipole MS and QBIC MS resonance, respectively. We intuitively compare the amplitude changes in these two modes under different biomolecule concentrations. Among them, the QBIC resonance shows a more significant amplitude change when detecting lower concentration samples, which is 16 times larger than the amplitude changes in the dipole resonance. The direct LoD of the QBIC MS sensor for arginine is 0.36 ng/mL. In order to ensure the reliability and accuracy of the experimental results, we measured each concentration three times and calculated the average value and standard deviation according to the three groups of measurement data of each concentration. In addition, we have also carried out sensing measurements on three identical samples, and the results of three groups of experiments show that the amplitude variation trend of transmission spectrum is consistent, and the error of sensing amplitude variation is about 10%. In order to further explain the mechanism of the resonance amplitude change in the QBIC MS biosensor, we simulate the effect of the dielectric loss of the microorganism to be tested on the transmission spectra of the proposed biosensor, and the loss tangent (tan δ) is the dielectric loss of the microorganism to be measured in the simulation. We fix the emissivity of the object under test to 1.5 and the thickness to 1 μm. When the loss tangent increases from 0 to 0.2, the resonance peak transmission amplitude significantly decreases from 0.86 to 0.93, as shown in Figure 7c,f, respectively. It can be concluded that dielectric loss is a sensitive factor affecting the size of the biosensor transmission peak. This capability provides a new degree of freedom for biomolecule detection applications.

By referring to the literature, Table 1 presents a comparison between the biosensor proposed in this work and other terahertz biosensors reported recently [24,48,49,50,51,52]. Here, we mainly focused on the resonator materials and the limit of detection (LoD) of different biosensors. It is seen from the table below that the LoD of our biosensor is much better than that of other work. The sensor we designed in this study has higher sensitivity sensing performance and higher cost performance.

## 4. Conclusions

A QBIC MS with a high Q-factor was designed and fabricated by introducing asymmetry degree α into the unit structure, and the characteristics and feasibility of the proposed QBIC MS were investigated systematically via numerical simulations, theoretical study, and experimental test. The QBIC MS sensor was applied to the detection of arginine molecules. The THz amplitude transmission of the QBIC MS changes significantly under different concentrations of arginine, and the LoD for arginine molecules reaches 0.36 ng/mL. Experimental results show that the QBIC MS sensor is more sensitive to biomolecules than the traditional dipole resonance sensor. In the future, we hope to design QBIC MS biosensors with specific frequencies to achieve specific detection of biomolecules, and the actual performance of the device can be improved.

## Figures and Tables

**Figure 1 nanomaterials-14-00799-f001:**
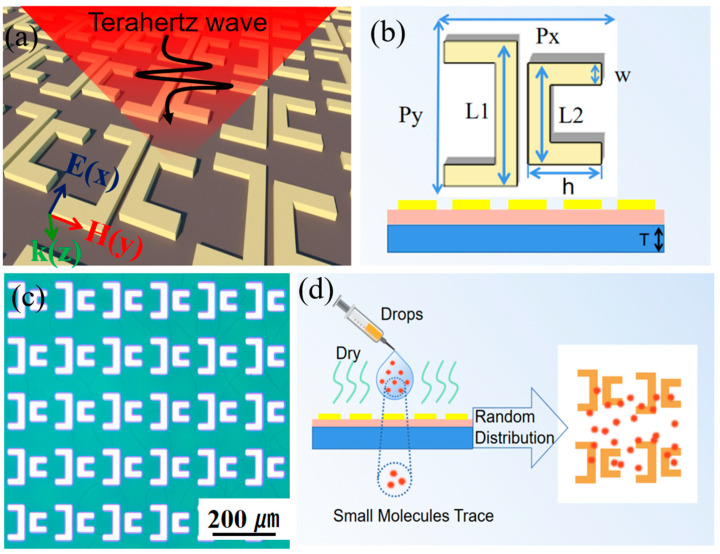
(**a**) Schematic diagram of metal QBIC MS for biosensors. (**b**) The parameters of one unit in (**a**). (**c**) The optical image of the biosensor structure. (**d**) The schematic diagram for sensing process based on QBIC MS.

**Figure 2 nanomaterials-14-00799-f002:**
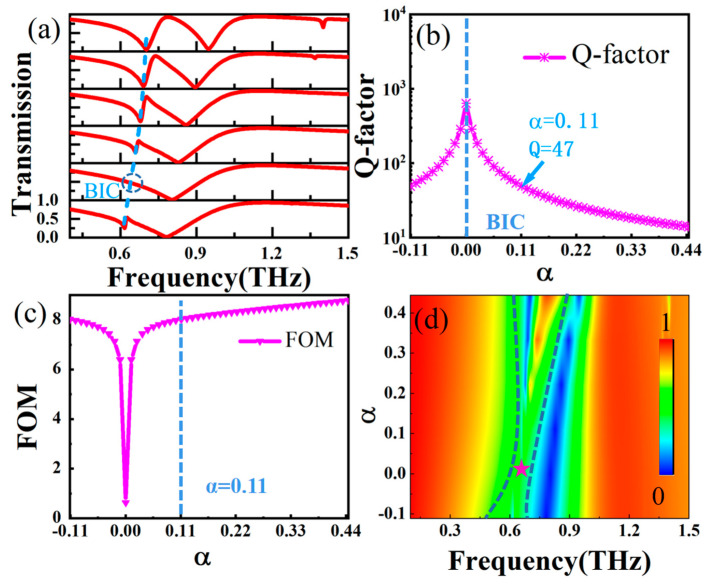
(**a**) THz transmission spectra with α = −0.11, 0, 0.11, 0.22, 0.33, 0.44. (**b**) Q-factor as a function of asymmetry degree, α calculated from THz transmission spectra. (**c**) FoM of QBIC structure as a function of asymmetry degree α. (**d**) Two-dimensional plots of THz transmission with respect to frequency and asymmetry degrees α.

**Figure 3 nanomaterials-14-00799-f003:**
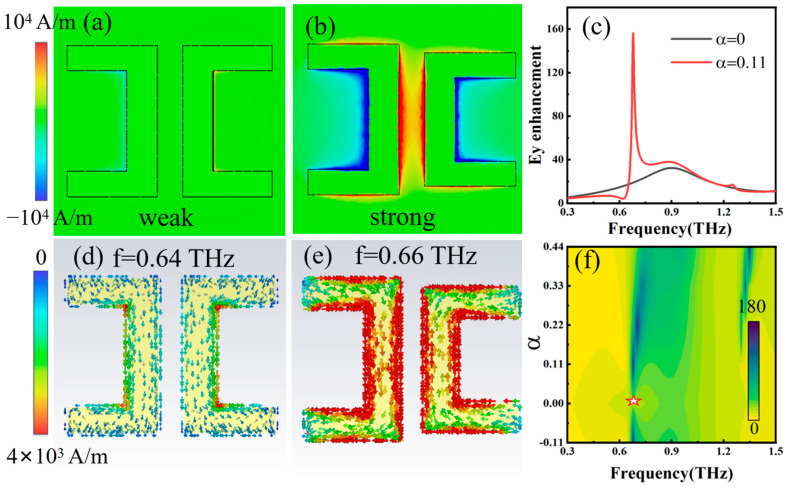
Magnetic field intensity diagram along the z direction at 0.64 THz (**a**) and 0.66 THz (**b**). Surface current diagram at 0.64 THz (**d**) and 0.66 THz (**e**). The 0.64 THz and 0.66 THz correspond to the frequencies of BIC and QBIC states with α = 0 and α = 0.11, respectively. (**c**) Electric field enhancement of BIC response (α = 0) and QBIC response (α = 0.11). (**f**) Color plots of electric field enhancement with respect to α and frequency; the pink state indicates the BIC resonant mode.

**Figure 4 nanomaterials-14-00799-f004:**
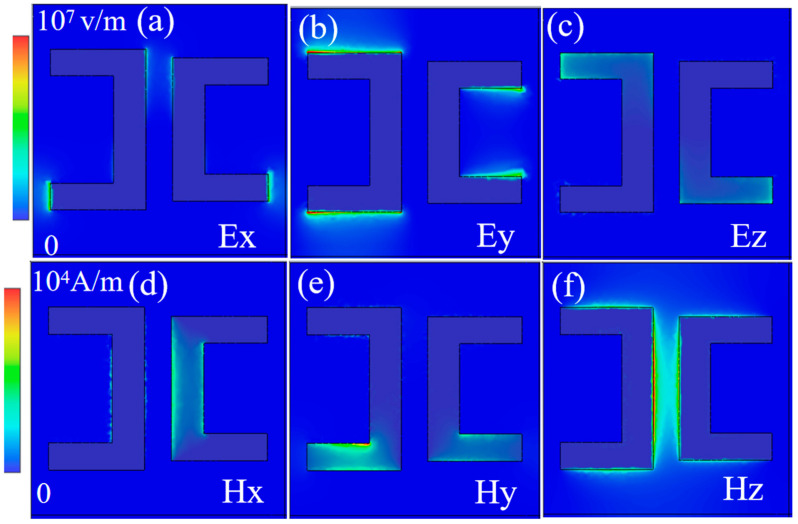
The distribution of electric field components (**a**–**c**) and magnetic field components (**d**–**f**) at 0.66 THz (QBIC position).

**Figure 5 nanomaterials-14-00799-f005:**
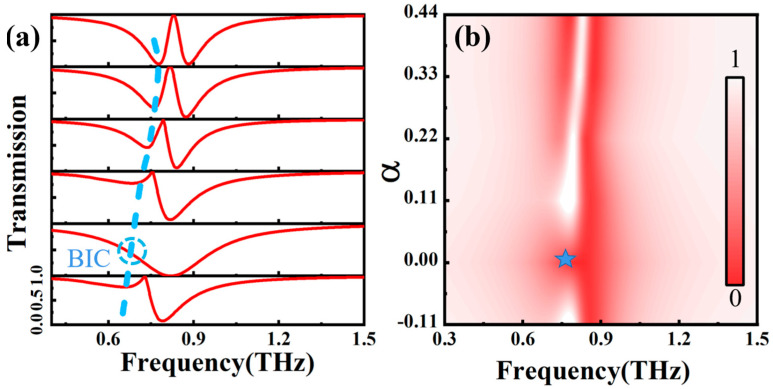
(**a**) Calculated THz transmission spectra of the QBIC MS for various α with coupled resonator model. The BIC position is marked with a blue circle. (**b**) Two-dimensional plots of THz transmission with respect to frequency and asymmetry degrees α.

**Figure 6 nanomaterials-14-00799-f006:**
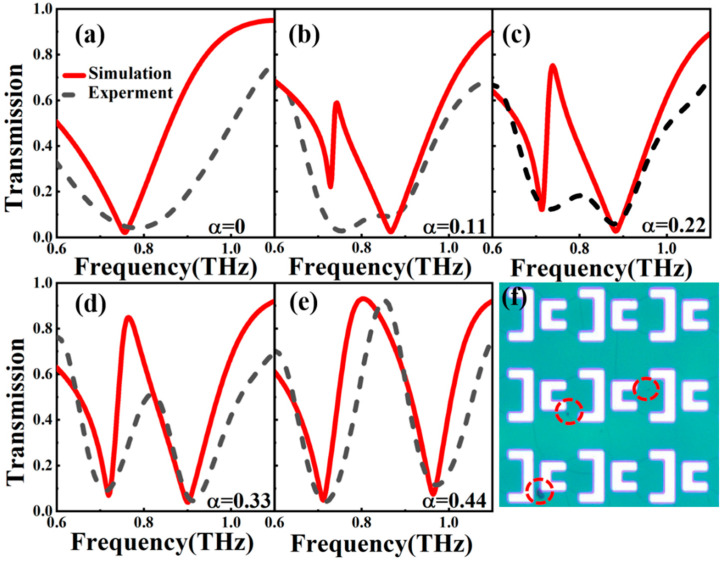
Transmission spectra of THz MSs with different α (α=(L1−L2)/L1), α = 0 (**a**), 0.11 (**b**), 0.22 (**c**), 0.33 (**d**), and 0.44 (**e**). The black dashed line is the experimental data, and the red solid line is from the simulation. (**f**) Photo image of a QBIC MS array; the red circles mark some impurities that formed during the manufacturing process.

**Figure 7 nanomaterials-14-00799-f007:**
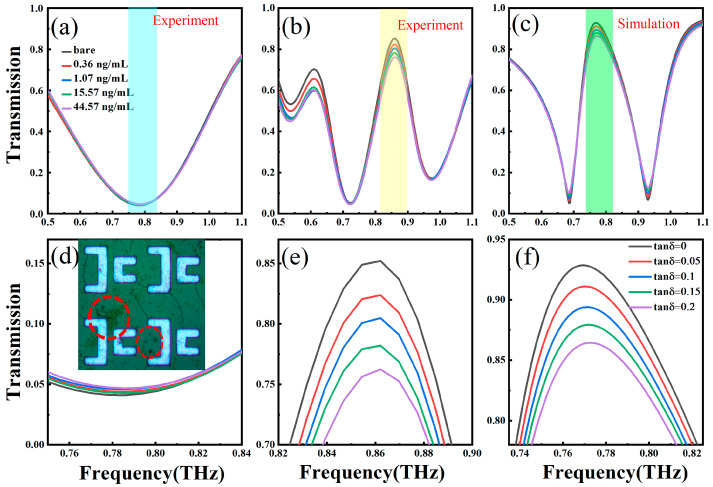
Experimental transmission spectra of arginine biomolecules with various concentrations on classical dipole MSs (**a**) and QBIC MSs (**b**). (**c**) Simulation results in which the dependence of transmittance on the analyte loss tangent increases from 0 to 0.2. The magnified views of the amplitude changes in the classical dipole MS (**d**) and QBIC MS (**e**) resonances. (**f**). The magnified views of the amplitude changes in simulated QBIC MS. The inset in (**d**) is a photomicrograph of the biosensor surface at a sample molecular concentration of 44.57 ng/mL. Sample molecules freely distributed in the form of small clusters are marked with red circles.

**Table 1 nanomaterials-14-00799-t001:** Comparison among different THz sensors.

Resonator Material	Resonance Type	Sensing Type	Analyte	LOD	Ref.
Carbon nanotube	LC resonance	Amplitude	Glucose	20 ng/mL	[48]
Al + Au nanoparticles	Toroidal dipole response	Frequency	Hemagglutinin tag protein	20 ug/mL	[24]
Au	LSP resonance	Frequency	Toxaphene	213 mg/mL	[49]
Al + graphene + perovskite	EIT-like resonance	Frequency, phase, amplitude	Whey protein	6.25 ng/mL	[50]
Au + graphene	Dipolar-like response	Amplitude	Doxycycline hydrochloride	10 mg/mL	[51]
Graphene	Waveguide response	Amplitude	Chlorpyrifos methyl	0.13 mg/mL	[52]
Al	BIC resonance	Amplitude	Arginine	0.36 ng/mL	This work

## Data Availability

Data available in a publicly accessible repository.

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
