# Peer review of "Terahertz Biosensor Engineering Based on Quasi-BIC Metasurface with Ultrasensitive Detection"

_nanomaterials, 2024, doi:10.3390/nano14090799_

Round 1

Reviewer 1 Report

Comments and Suggestions for Authors

The authors proposed a double C-shaped metasurface that supports quasi-bound state in the continuum  (QBIC) resonances as well as proved that the designed QBIC metamaterial shows very high sensitivity for detecting concentrations of arginine biomolecules: the amplitude of the resonance based on the QBIC mode variation is 16 times larger than that of the classical dipole resonance. As a result of an in-depth review, I conclude that the manuscript is interesting, but it lacks a broader discussion regarding the following issues:

1.     In the introduction, the authors mention the emergence of metamaterials. It is worth expanding this context by mentioning the wider range and possibilities of THz metamaterials, e.g. Liquid Crystals 39.6 (2012): 739-744; Optics letters 40.13 (2015): 3197-3200, etc.

2.     The description of the simulation is too general and makes it difficult to reproduce the results. Did the authors take into account the dispersion of individual material parameters? What were the boundary conditions, the size of the numerical grid, the distance of the ports from the structure?

3.     Will simulating a metal based solely on conductivity fully reflect its properties in experimental studies? In other words, what simplification does this result in?

4.     Why does the accuracy of reproducing the experimental results increase as the value of parameter α increases (Fig. 5)?

5.     I encourage the authors to insert a table comparing the most important parameters of the presented THz metamaterial to other similar THz meta-devices for the detection of microorganisms. This will add value to the manuscript.

Author Response

please refer to the response letter attached

Reviewer 2 Report

Comments and Suggestions for Authors

In this paper, authors considered the Terahertz Biosensor Engineering Based on Quasi-BIC Metasurface with Ultrasensitive Detection. In my opinion, the results look very interesting and important for biological applications. But the interpretation of results should be improved. The BIC phenomenon is result of excitation of trapped mode. The authors should add other fields components (Ex,Ey,Ez and Hx,Hy,Hz). In order to confirm trapped mode, the z?! component of electric or magnetic?! field will be dominated. Please add this analysis. 

After that, the paper can be published.

Author Response

(The authors gave the same response as above.)

Reviewer 3 Report

Comments and Suggestions for Authors

The authors present interesting studies on a Q-BIC metasurface engineered to function as a high-quality sensor. I can recommend publication after the following observations are addressed:

 1. The authors show no results in the sensitivity of the sensor with respect to external factors.

2. The spectral response in Figure 6(b) shows the peak between 0.8 and 0.9 THz being chosen, as opposed to the one close to 0.6 THz. The authors should explain why they made that choice. From the graph, the sensitivity of the Q-BIC sensor seems to be higher than the one chosen.

The authors may want to improve the introduction by adding the following literature: https://doi.org/10.3390/polym15030545 and https://doi.org/10.3390/s21165600, which describe a similar type of sensor.

Based on observations 1 and 2 above, I recommend the authors perform a Major Revision, in which a study on sensitivity is performed for the spectral response at the selected peaks.

Comments on the Quality of English Language

No comments

Author Response

(The authors gave the same response as above.)

Round 2

Reviewer 3 Report

Comments and Suggestions for Authors

The authors have responded to all my observations. I recommend publication.

Comments on the Quality of English Language

No comments